# Knowledge, attitudes, and practices regarding Crimean-Congo hemorrhagic fever among general people: A cross-sectional study in Pakistan

Hashaam Jamil[1], Muhammad Fazal Ud Din[1], Muhammad Junaid Tahir[1], Muhammad Saqlain[2], Zair Hassan[3], Muhammad Arslan Khan[4], Mustafa Sajjad Cheema[5], Irfan Ullah[6], Md. Saiful Islam[7,8]*, Ali Ahmed[9]

1 Department of Medicine, Lahore General Hospital, Lahore, Pakistan, 2 Department of Pharmacy, Quaid-I-Azam University, Islamabad, Pakistan, 3 Department of Cardiology, Lady Reading Hospital, Peshawar, Pakistan, 4 Department of Pharmaceutical Sciences, University of Lahore Teaching Hospital, Lahore, Pakistan, 5 Department of Medicine, CMH Lahore Medical College, Lahore, Pakistan, 6 Department of Medicine, Kabir Medical College, Gandhara University, Peshawar, Pakistan, 7 Department of Public Health and Informatics, Jahangirnagar University, Dhaka, Bangladesh, 8 Centre for Advanced Research Excellence in Public Health, Dhaka, Bangladesh, 9 Department of Pharmacy, School of Pharmacy, Monash University, Selangor, Malaysia

* islam.msaiful@outlook.com

## Abstract

### Background

Crimean-Congo hemorrhagic fever (CCHF) continues to pose a serious threat to the fragile healthcare system of Pakistan with a continuous increase of morbidity and mortality. The present study aimed to assess the knowledge, attitudes, and practices regarding CCHF among general people who resided in Pakistan.

### Methods

An online cross-sectional survey design was applied, and a convenience sampling technique was used to recruit 1039 adult people from Pakistan. Data were collected from September 08 to October 12, 2021. The questionnaire consisted of a total of 32 questions in four parts assessing socio-demographics, as well as knowledge, attitudes, and practices regarding CCHF. All statistical analyses were performed using the Statistical Package for the Social Sciences (SPSS), and logistic regression analyses were performed to determine the factors associated with good knowledge, positive attitudes, and good practices.

### Results

Alarmingly, 51.5% of participants heard about CCHF infection before administering the survey. Among these, 20.2%, 33.3%, and 48.2% of the study participants had demonstrated good knowledge, positive attitudes, and good practices, respectively. Binary logistic regression analysis revealed that education and income status had a significant impact on

**Data Availability Statement:** All relevant data are within the manuscript and its Supporting Information files.

**Funding:** The authors received no specific funding for this work.

**Competing interests:** The authors have declared that no competing interests exist.

knowledge and attitudes ($p<0.05$). Similarly, the mean attitude scores differed significantly by age, education, and income status ($p<0.05$).

## Conclusions

The findings reflected inadequate levels of knowledge, attitudes, and practices regarding CCHF among general people in Pakistan which may regard as lower than expected. As CCHF is a highly contagious disease, it's urgent to initiate a comprehensive approach to handle the situation before it spreads further in Pakistan.

## Author summary

Crimean-Congo hemorrhagic fever (CCHF) is usually transmitted by ticks or contact with viremic animal tissues (animal tissue where the virus has entered the bloodstream) during and immediately after post-slaughter of animals and can lead to epidemics, has a high case fatality ratio (10–40%), potentially results in hospital and health facility outbreaks, and is difficult to prevent and treat. The number of CCHF infections increases around Eid-ul-Adha as more livestock is moved from rural areas to cities during Eid-ul-Adha. In Pakistan, the current national control program has been unable to eliminate CCHF on its own. Therefore, sociocultural and behavioral research can inform and improve the impact of future control programs. To this end, we investigated knowledge, attitudes, and practices related to CCHF in general population of Pakistan. We alarmingly found low levels of disease knowledge and attitudes and practices. Preventive interventions are uncommon due to poor infrastructure, a lack of education, and restricted access to health care and livestock-related facilities. It is high time that Pakistan's health, agriculture, and media sectors collaborate with international organizations to establish and implement a strategic framework for CCHF awareness and prevention. This kind of social context is vital to future public health campaigns, and highlights the importance of cross-disciplinary work to achieve successful disease control.

## Background

Crimean-Congo hemorrhagic fever (CCHF) is a zoonotic viral disease caused by the Crimean-Congo hemorrhagic fever virus (CCHFV) [1]. CCHFV belongs to the genus *Orthonairovirus* of the family *Bunyaviridae*. The virus is transmitted to humans mostly by the bites of infected ticks or via contact with the secretions and body fluids of the infected animals [2]. The most common vector for this arthropod-borne disease is hard ticks of the *Ioxididae* family especially those of the genus *Hyalomma marginatum*. Human-to-human transmission can also occur through direct contact with tissues and body fluids of the infected people, especially in healthcare settings. Various wild and domestic animals serve as reservoirs of this deadly disease, including sheep, goats, cattle and hares, etc. [3]. Consumption of the under-cooked meat of infected animals has also been reported to be responsible for the transmission of CCHFV to humans [4].

The incubation period varies depending upon several factors, including the mode of acquisition of the virus. The incubation period following a tick bite ranges from 2 to 7 days. However, it is usually 10–14 days following contact with infected body fluids or tissues [5]. Nosocomial infections appear to have an even shorter interval between the contact and

appearance of symptoms [6]. Following the incubation period, nonspecific febrile symptoms overlap with other viral hemorrhagic illnesses. Common symptoms of this viral disease include sudden fever, headaches, fatigue, myalgia, abdominal pain, ecchymosis, and petechial hemorrhages. Some patients also present with extensive hemorrhages, hepatic dysfunction, and other gastrointestinal symptoms [7, 8]. Since the clinical features and early laboratory findings very much overlap with other viral hemorrhagic diseases, the definitive diagnosis is only made based on specific tests. In the early phase of the viral illness, diagnosis can be made by detecting viral nucleic acid by using reverse transcription-polymerase chain reaction (RT-PCR), while in the late stages, it can be confirmed by detecting antibodies to the specific antigen [9]. The case fatality rate of CCHF is 10–40% [10]. Although ribavirin is used in severe cases, no effective antiviral therapy for CCHFV is present, and treatment is mainly supportive. Two vaccines have been developed but are not currently recommended for public administration [8].

CCHF was first recognized in the Crimean region of the former Soviet Union in 1944, and the virus was first isolated in 1969 in the Democratic Republic of Congo, thus resulting in the current name of the disease [11]. Since its discovery, nearly 140 outbreaks involving more than 5,000 cases have been reported all over the world. A total of 52 countries have been recognized as endemic, reporting significant number of cases every year [12]. CCHFV is responsible for outbreaks in many areas of the Middle East, Europe, Asia, and Africa [13]. The five countries currently having the strongest evidence for presence of CCHF are Turkey, Iran, Afghanistan, Tajikistan, and Pakistan [14]. In Pakistan, the first confirmed case of Congo fever was reported in 1976 at Rawalpindi. After that, a number of cases have been reported throughout the country. According to the national institute of health (NIH), 365 confirmed cases of CCHF were reported between the years of 2014 and 2020 with a 25% fatality rate [15–17]. In July 2014, an outbreak in Hayatabad Medical Complex (HMC) resulted in the deaths of eight patients including one nurse [18]. Another outbreak in May 2017 was reported in the Karak district of Khyber Pakhtunkhwa in which six patients presented with nausea, vomiting, and diarrhea. Two of these patients later developed a severe bleeding disorder. This outbreak ultimately resulted in the deaths of four of these six patients within four days [19]. In 2021, NIH reported 14 confirmed cases and 5 deaths in Balochistan [20]. During Eid-ul-Adha, the largest Muslim religious festival in which millions of animals are slaughtered, the rate of new infections increases several times. A study showed that most of the cases reported in Pakistan were seen around the Eid-ul-Adha season [21]. This trend may be explained by the movement of a more livestock from rural areas to the cities during Eid-ul-Adha.

CCHF continues to pose a serious threat to the fragile healthcare system of Pakistan with a continuous increase in cases in the past decade. Knowledge, attitudes, and practices (KAP) studies are critical especially in public health, in assessing information on current programs, formulating behavioral strategies, and implementing new public health programs [22]. Some studies have been conducted to show KAP of the population at risk, i.e., butchers, healthcare professionals, people living in rural areas, etc., involving certain areas of Pakistan [23–25]. This present study aimed to assess the knowledge, attitudes, and practices regarding CCHF among general people in Pakistan. This study's findings will help policymakers develop strategies and interventions to raise awareness among general people and prevent future outbreaks.

## Methods

### Ethics statement

The study ensured that the privacy of each participant was adequately protected. The study did not contain any names or emails so that the participant could not be tracked. Participants were

allowed to withhold the completed form at any time before submitting it. The study protocol was approved by the ethical review committee of a Medical Teaching Institution, Lady Reading Hospital Peshawar, Pakistan (Ref. No: 710/LRH/MTI). A written consent from the participant after being informed about the purpose of the study and research objectives was obtained at the start of the online survey.

## Study design and setting

A convenient sampling technique was used to conduct a cross-sectional survey among general people in Pakistan. We adopted convenient sampling because it is extremely speedy, easy, readily available, and cost-effective. Although there are biases in this sampling technique, we covered them by taking a larger sample size and inviting participants from all types of general populations. A semi-structured questionnaire including informed consent was incorporated in the Google Forms, and a shareable link was then created. Data were collected from September 08 to October 12, 2021 in the four provinces of Pakistan (i.e., Sindh, Punjab, Balochistan, and Khyber Pakhtunkhwa).

## Sample size

The larger the target sample size, the higher the external validity of the study. This study aimed to maximize reach and gather data from as many respondents as possible. According to the latest United Nations data, Pakistan has a population of 230,275,648 [26]. The representative target sample size needed, to achieve the study objectives and sufficient statistical power, was calculated through an online calculator (i.e., Raosoft) [27].

The sample size was calculated as 752 participants, using a margin of error of 3%, a confidence level of 90%, a 50% response distribution, and people 230,275,648. To get more reliable results, the total number of participants in this study was 1039.

## Questionnaire development

The questionnaire was adopted through an intensive review of the literature [2, 3, 28] and reviewed by the research committee comprised of senior epidemiologists and physicians having relevant research experiences. After discussion and review, the authors finalized the questionnaire. Then, a pilot study was conducted with 80 participants to check the reliability of the questionnaire where the Cronbach alpha of knowledge, attitudes, and practices was 0.76 which is well accepted in its conventional thresholds. Finally, the questionnaire was distributed among participants for collecting their responses. The questionnaire was comprised of an introductory paragraph, clarifying the aim and objectives of the study; followed by mandatory informed consent for all participants; and then four sections assessing socio-demographics, knowledge, attitudes, and practices.

## Socio-demographic information

The socio-demographic section consisted of six questions including gender, age, marital status, education, residence, and monthly income. After the socio-demographic section, there is a single question, *"Have you heard about Congo fever before?"*. For all those participants who responded *"No"* option, their forms were submitted automatically after the socio-demographic section, and only those who responded *"Yes"*, were allowed to survey in the next three sections.

### Knowledge regarding CCHF

The knowledge section consisted of twelve questions. The correct answer was coded as 1, while the wrong answer was coded as 0. The total score was obtained by summating the raw score of each item and ranged from 0 to 12, with an overall greater score indicating more accurate knowledge. A cut-off level of $\geq 9$ was set for categorizing the good knowledge.

### Attitudes towards CCHF

The attitude section consisted of nine questions, and the response of each question was indicated on a 5-point Likert scale as follows:1 ("*Strongly disagree*"), 2 ("*Disagree*"), 3 ("*neutral*"), 4 ("*Agree*"), and 5 ("*Strongly agree*"). The total score was calculated by summating the raw scores of the nine questions ranging from 9 to 45, with an overall higher score indicating more positive attitudes. A cut-off level of $\geq 34$ was set for a more positive attitude.

### Practices regarding CCHF

The practice section consisted of five questions, and the response to each question was indicated on a 5-point Likert scale as follows: 1 ("*Strongly disagree*"), 2 ("*Disagree*"), 3 ("*neutral*"), 4 ("*Agree*"), and 5 ("*Strongly agree*"). The total score was calculated by summating the raw scores of the five questions ranging from 5 to 25, with an overall higher score indicating the good practices toward CCHF. A score of $\geq 19$ was calibrated as good practices toward CCHF.

### Data collection and sampling

All individuals, aged $\geq 18$ years of either gender (male or female), living in Pakistan were eligible to participate in the survey. Those participants, who refused to provide informed consent were excluded from the study. The questionnaire was designed in two languages, one in the native language of the population, that is, Urdu, and the other in English. Data was collected through friend circle forwarding, WhatsApp sharing, and other social media platforms (e.g., Facebook, Gmail, etc.). The survey was completely voluntary, and participants could withdraw their responses from the survey at any moment as per their choice. Incomplete submission of the survey questionnaire was not possible due to the feature in the Google Forms that prevented the submission of partially answered or partially filled questions.

### Statistical analysis

All statistical analyses were performed using the Statistical Package for the Social Sciences (SPSS) version 21. Inferential statistics were used, depending on the nature of the data and the variables. Logistic regression analyses were performed to find the factors of good knowledge, positive attitudes, and good practices. Results were expressed as odds ratio (OR) with a 95% confidence interval (CI), and *p*-value. A *p*-value of less than 0.05 was considered statistically significant.

## Results

### Socio-demographic characteristics

A total of 1039 participants were included in the final analysis. Of them, the majority were male (51.4%), had ages ranging from 22–25 years (38.9%), and came from urban areas (76.3%). Additionally, 55.6% had a graduation level of education and followed by a post-graduation level (18.1%). Most reported themselves as unmarried (79.5%) (Table 1). Among those

**Table 1. Study samples' socio-demographic characteristic (N = 1039).**

| Variables | Frequency | Percentage |
|---|---|---|
| **Gender** | | |
| Male | 534 | 51.4 |
| Female | 505 | 48.6 |
| **Age** | | |
| 18–21 | 347 | 33.4 |
| 22–25 | 404 | 38.9 |
| 26 < | 288 | 27.7 |
| **Marital Status** | | |
| Married | 213 | 20.5 |
| Unmarried | 826 | 79.5 |
| **Residence** | | |
| Urban | 793 | 76.3 |
| Rural | 246 | 23.7 |
| **Education** | | |
| Primary or below | 20 | 1.9 |
| Matriculation | 72 | 7.0 |
| Intermediate | 181 | 17.4 |
| Graduation | 578 | 55.6 |
| Post-graduation | 188 | 18.1 |
| **Monthly income** | | |
| <25000 | 282 | 27.1 |
| 25000–50000 | 371 | 35.7 |
| >50000 | 386 | 37.2 |
| **Have you heard about Congo fever?** | | |
| Yes | 535 | 51.5 |
| No | 504 | 48.5 |

participants who had heard about CCHF, the majority were male (56.2%), unmarried (79.5%), from urban areas (80.7%), and had a graduation level of education (57.4%) (Table 2).

## General knowledge about CCHF

Alarmingly, 48.5% (n = 504) respondents had never heard about CCHF infection before administering the survey. Data were analyzed from those participants, who had heard about CCHF (n = 535,51.5%). Among these 535 participants, 27.1% (n = 145) knew that CCHF was first characterized in Crimean and 71% (n = 380) participants knew the common symptoms of CCHF are sudden fever, headache, and myalgias. 70.5% (n = 377) believe that CCHF is completely curable, and 58.9% (n = 315) thought that vaccine is available for CCHF. Overall, 20.2% (n = 108) participants have good knowledge of CCHF, while 79.8% (n = 427) have poor knowledge (Table 3).

## Attitudes and practices

Only 33.3% (n = 178) of the participants had overall good attitudes. Mostly, 49.9% (n = 267) agreed that CCHF is a dangerous disease, but at the same time, 29.5% (n = 158) of participants disagreed on whether they are at risk of getting the CCHF. 42.4% (n = 227) of respondents agreed that there is an increased risk of people getting the disease during Eid-ul-Adha. When asked about spiritual healers/transitional healers who can treat CCHF completely, 30.1% (n = 161) were neutral to making the decision (Table 4).

**Table 2. Socio-demographic of participants, who had heard about CCHF (*N* = 535).**

| Variables | Frequency | Percentage |
|---|---|---|
| **Gender** | | |
| Male | 301 | 56.2 |
| Female | 234 | 43.7 |
| **Age** | | |
| 18–21 | 158 | 29.5 |
| 22–25 | 218 | 40.7 |
| 26 < | 159 | 29.8 |
| **Marital Status** | | |
| Married | 110 | 20.5 |
| Unmarried | 425 | 79.5 |
| **Residence** | | |
| Urban | 432 | 80.7 |
| Rural | 103 | 19.3 |
| **Education** | | |
| Primary or below | 5 | 0.9 |
| Matriculation | 26 | 4.9 |
| Intermediate | 91 | 17.0 |
| Graduation | 307 | 57.4 |
| Post-graduation | 106 | 19.8 |
| **Monthly income** | | |
| <25000 | 119 | 22.2 |
| 25000–50000 | 185 | 34.6 |
| >50000 | 231 | 43.2 |

Assessment of practice showed that 48.2% (n = 258) participants had a good practice. 51.6% (n = 276) of participants agreed that doctors and other medical professionals can provide accurate information about CCHF. 40.9% (n = 219) of participants strongly agreed that there is a need for more awareness of CCHF in the general public about CCHF (Table 5).

## Factors of good knowledge, attitudes, and practices

Binary logistic regression analysis revealed that education had significant impact on knowledge regarding CCHF as participants having matriculation degrees had lower odds compared to those who had post-graduation degree (OR = 0.22; 95% CI = 0.05–0.99, *p* = 0.049). Participants who had intermediate education were 0.48 times less likely to have good knowledge compared to those had post-graduation degree (OR = 0.48; 95% CI = 0.24–0.99, *p* = 0.045). The lower-income group were also less likely to have good knowledge compared to high-income group (OR = 0.26; 95% CI = 013–0.53, *p* < 0.001) (Table 6).

Based on the regression analysis, the lower age group (18–21 years) had lower odds of positive attitudes compared to participants with 26 years old or more (OR = 0.62; 95% CI = 0.38–0.99, *p* = 0.046). The participants who had matriculation (OR = 0.26; 95% CI = 0.08–0.80, *p* = 0.019) and intermediate education (OR = 0.45; 95% CI = 0.24–0.83, *p* = 0.011) were less likely to have positive attitudes than those had post-graduation degree. The income of participants also had significant effect on attitudes as lower-income group also showed lower odds of positive attitude compared to high-income groups (OR = 0.53; 95% CI = 0.32–0.86, *p* = 0.011) (Table 7).

**Table 3. Knowledge among general people regarding Congo (N = 535).**

| Variables | Frequency | Percentage |
|---|---|---|
| **Congo fever disease was first characterized in which area?** Correct "Crimean" | | |
| Incorrect | 390 | 72.9 |
| Correct | 145 | 27.1 |
| **Is CCHF transmissible?** Correct "Yes" | | |
| Incorrect | 148 | 27.7 |
| Correct | 387 | 72.3 |
| **The causative factor of CCHF is?** Correct "Virus" | | |
| Incorrect | 143 | 26.7 |
| Correct | 392 | 73.3 |
| **How CCHF Spreads to humans?** Correct" Infected tick bite, Contact with infected animal meat, secretions, and blood of infected humans" | | |
| Incorrect | 177 | 33.1 |
| Correct | 358 | 66.9 |
| **Highest number of cases of Congo virus are reported in which province?** Correct "Balochistan" | | |
| Incorrect | 353 | 66 |
| Correct | 182 | 34 |
| **Most suitable season for spread of CCHF is?** Correct "Summer" | | |
| Incorrect | 412 | 77 |
| Correct | 123 | 23 |
| **People at most risk for getting the disease are?** Correct "Livestock workers, slaughter house workers, and farmers" | | |
| Incorrect | 205 | 38.3 |
| Correct | 330 | 61.7 |
| **Common symptoms of CCHF are?** Correct "Sudden fever, headache, and myalgias" | | |
| Incorrect | 155 | 29 |
| Correct | 380 | 71 |
| **Do animals who have virus in their blood show any symptoms?** Correct "No" | | |
| Incorrect | 345 | 64.5 |
| Correct | 190 | 35.5 |
| **What are ways to prevent the spread of CCHF?** Correct "Screen animals for ticks, avoid contact, handling of blood and meat of infected animals" | | |
| Incorrect | 142 | 26.5 |
| Correct | 393 | 73.5 |
| **Is CCHF completely curable?** Correct "Yes" | | |
| Incorrect | 158 | 29.5 |
| Correct | 377 | 70.5 |
| **Is there any vaccine available for CHF?** Correct "No" | | |
| Incorrect | 220 | 41.1 |
| Correct | 315 | 58.9 |
| **Overall knowledge** | | |
| Poor (1–8) | 427 | 79.8 |
| Good (9–12) | 108 | 20.2 |

*Note*: Knowledge section was assessed by giving a score of 1 to correct answers and 0 to wrong answers. A score of greater than equal to 9 was regarded as good and a score of less than 9 was regarded as poor.

**Table 4. Attitude among general people regarding Congo (N = 535).**

| Variables | SD | D | N | A | SA |
|---|---|---|---|---|---|
| Do you believe that CCHF is a dangerous disease? | 39 (7.3) | 25 (4.7) | 65 (12.1) | 267 (49.9) | 139 (26.0) |
| Do you believe that you are at risk of getting the disease | 88 (16.4) | 156 (29.2) | 158 (29.5) | 111 (20.7) | 22 (4.1) |
| Do you believe that there is an increased risk of people getting the disease during Eid-ul-Adha? | 41 (7.7) | 53 (9.9) | 100 (18.7) | 227 (42.4) | 114 (21.3) |
| Do you believe that screening the animals for ticks and isolating the animals if they have ticks? | 35 (6.5) | 30 (5.6) | 56 (10.5) | 271 (50.7) | 143 (26.7) |
| Do you believe that CCHF can spread from an infected person to healthy person through skin? | 66 (12.3) | 206 (38.5) | 123 (23) | 104 (19.4) | 36 (6.7) |
| Do you believe that you can get CCHF from eating meat of infected animals? | 40 (7.5) | 75 (14.0) | 105 (19.6) | 244 (45.6) | 71 (13.3) |
| Suppose if someone in your family gets CCHF, do you think they should be admitted to a hospital or other healthcare facility | 39 (7.3) | 38 (7.1) | 64 (12) | 253 (47.3) | 141 (26.4) |
| Do you think there is a better chance to recover completely if infected person gets help from healthcare facility immediately? | 31 (5.8) | 24 (4.5) | 59 (11) | 265 (49.5) | 156 (29.2) |
| Do you believe that spiritual healers/transitional healers can treat CCHF completely? | 31 (5.8) | 86 (16.1) | 161 (30.1) | 153 (28.6) | 104 (19.4) |
| Overall attitudes | Negative (1–33) | | | 357 (66.7) | |
| | Positive (34–45) | | | 178 (33.3) | |

*Note*: SD = strongly disagree, D = Disagree, N = Neutral, A = Agree, SA = Strongly agree. The attitudes section was assessed by giving a score of 1 to strongly disagree and 5 to strongly agree A score of greater than or equal to 34 was regarded as positive and a score of less than 34 was regarded as negative.

Participants aged ranging from 22–25 years had low odds ratio of good practice than those were 26 years old or more (OR = 0.56; 95% CI = 0.37–0.84, $p$ = 0.005). Similar to knowledge and attitudes, practices were also influenced by the education and income status. Participants with matriculation degree and lower income had lower odds of good practice toward CCHF (OR = 0.38; 95% CI = 0.15–0.95, $p$ = 0.040, and OR = 0.40; 95% CI = 0.25–0.64, $p$ < 0.001, respectively) (Table 8).

**Table 5. Practices among general people regarding Congo (N = 535).**

| Variables | SD | D | N | A | SA |
|---|---|---|---|---|---|
| Do you trust doctors and other medical professionals can provide you with accurate information about Congo fever? | 46 (8.6) | 24 (4.5) | 78 (14.6) | 276 (51.6) | 111 (20.7) |
| Do you take standard precautions during handling of animals and Qurbani during Eid-ul-Adha? | 31 (5.8) | 59 (11) | 90 (16.8) | 260 (48.6) | 95 (17.8) |
| Do you think keeping livestock at home poses you an extra risk of getting CCHF | 32 (6.0) | 56 (10.5) | 116 (21.7) | 262 (49.0) | 69 (12.9) |
| Do you think there is a need of more awareness of CCHF in general public about CCHF | 34 (6.4) | 20 (3.7) | 65 (12.1) | 197 (36.8) | 219 (40.9) |
| Do you think hospitals in your area are provided with sufficient facilities to diagnose and treat CCHF | 28 (5.2) | 118 (22.1) | 169 (31.6) | 138 (25.8) | 82 (15.3) |
| Overall practice | Poor (1–18) | | | 277 (51.8) | |
| | Good (19–25) | | | 258 (48.2) | |

*Note*: SD = strongly disagree, D = Disagree, N = Neutral, A = Agree, SA = Strongly agree. The practices section was assessed by giving a score of 1 to strongly disagree and 5 to strongly agree A score of greater than or equal to 19 was regarded as positive and a score of less than 19 was regarded as negative.

**Table 6. Binary logistic regression analysis to find factors of good knowledge (*N* = 535).**

| Variables | Knowledge | | OR | (95% CI) | *p*-value |
|---|---|---|---|---|---|
| | Poor | Good | | | |
| **Gender** | | | | | |
| Male | 249 (82.7) | 52 (17.3) | 0.664 | 0.44–1.01 | 0.058 |
| Female | 178 (76.1) | 56 (23.9) | Reference | | |
| **Age** | | | | | |
| 18–21 | 129 (81.6) | 28 (18.4) | 0.753 | 0.44–1.29 | 0.309 |
| 22–25 | 176 (80.7) | 42 (19.3) | 0.809 | 0.49–1.33 | 0.404 |
| 26 > | 122 (76.7) | 37 (23.3) | Reference | | |
| **Marital status** | | | | | |
| Married | 93 (84.5) | 17 (15.5) | 0.671 | 0.38–1.18 | 0.167 |
| Unmarried | 334 (78.6) | 91 (21.4) | Reference | | |
| **Residence** | | | | | |
| Urban | 338 (78.2) | 94 (21.8) | 1.768 | 0.96–3.25 | 0.066 |
| Rural | 89 (86.4) | 14 (13.6) | Reference | | |
| **Education** | | | | | |
| Primary or below | 2 (40.0) | 3 (60.0) | 0.139 | 0.031–0.896 | 0.052 |
| Matriculation | 4 (92.3) | 2 (7.7) | 0.221 | 0.049–0.996 | **0.049** |
| Intermediate | 77 (84.6) | 14 (15.4) | 0.483 | 0.237–0.984 | **0.045** |
| Graduation | 245 (79.5) | 63 (20.5) | 0.683 | 0.410–1.136 | 0.142 |
| Post-graduation | 77 (72.6) | 29 (27.4) | Reference | | |
| **Monthly income** | | | | | |
| <25000 | 109 (91.6) | 10 (8.4) | 0.261 | 0.13–0.53 | **<0.001** |
| 25000–50000 | 147 (79.5) | 38 (20.5) | 0.737 | 0.46–1.17 | 0.195 |
| >50000 | 171 (74.0) | 60 (6.0) | Reference | | |

*Note*: OR = Odds Ratio, CI = Confidence interval.

A *p*-value of less than 0.05 was considered statistically significant.

## Discussion

To the best of our knowledge, this was the first nationwide study to investigate the knowledge, attitudes, and practice among general people residing in Pakistan. Previous studies in Pakistan had a small number of participants or were restricted to healthcare workers [28–30]. Our study revealed that Pakistani people had poor levels of knowledge, attitudes, and practices towards the CCHF. It's worth noting that about half of the study participants (48.5%) had never heard of the term "Congo", similar to a study which was conducted at three universities in Sindh, where 50.4% of participants had never heard about "Congo" [31]. Similarly, a survey of the general community of Rawalpindi revealed that 37% of the population was unaware of the term "Congo" [30].

Knowledge of the disease is considered the first stepping stone to any health education activity that is implemented. Knowing the causes and transmission sources of a disease, increases the likelihood that people will become more aware of the spread of communicable diseases, and of the preventive measures to slow transmission. In the present study, among those participants who had heard about CCHF, 79.8% of participants have poor knowledge about CCHF. 72.9% of participants did not know CCHF was first characterized in Crimean. 66% of participants were unaware that most cases of CCHF are reported in Baluchistan. At least 14 sporadic outbreaks have been reported in Pakistan since the year 2000, with nine outbreaks in the Balochistan province [32], where the majority of the people are illiterate. The bulk of them worked as shepherds and had insufficient knowledge, attitudes, and practices in

**Table 7. Binary logistic regression analysis to find factors of positive attitudes (N = 535).**

| Variables | Attitude | | Odds ratio | (95% CI) | p-value |
|---|---|---|---|---|---|
| | Negative | Positive | | | |
| **Gender** | | | | | |
| Male | 208 (69.1) | 93 (30.9) | 0.784 | 0.54–1.13 | 0.187 |
| Female | 149 (63.7) | 85 (36.3) | Reference | | |
| **Age** | | | | | |
| 18–21 | 115 (72.8) | 43 (27.2) | 0.617 | 0.38–0.99 | **0.046** |
| 22–25 | 143 (65.6) | 75 (34.4) | 0.865 | 0.57–1.32 | 0.505 |
| 26 > | 99 (62.3) | 60 (37.7) | Reference | | |
| **Marital Status** | | | | | |
| Married | 76 (69.1) | 34 (30.9) | 0.873 | 0.56–1.37 | 0.556 |
| Unmarried | 281 (66.1) | 144 (33.9) | Reference | | |
| **Residence** | | | | | |
| Urban | 286 (66.2) | 146 (33.8) | 1.13 | 0.71–1.79 | 0.598 |
| Rural | 71 (68.9) | 32 (31.1) | Reference | | |
| **Education** | | | | | |
| Primary or below | 4 (99) | 1 (1) | 0.251 | 0.076–0.684 | 0.159 |
| Matriculation | 22 (84.6) | 4 (15.4) | 0.256 | 0.082–0.796 | **0.019** |
| Intermediate | 69 (75.8) | 22 (24.2) | 0.449 | 0.243–0.832 | **0.011** |
| Graduation | 200 (69.4) | 107 (35.1) | 0.761 | 0.48–1.20 | 0.236 |
| Post-graduation | 62 (58.5) | 44 (41.5) | Reference | | |
| **Monthly income** | | | | | |
| <25000 | 89 (74.8) | 30 (25.2) | 0.528 | 0.32–0.86 | **0.011** |
| 25000–50000 | 127 (68.6) | 58 (31.4) | 0.715 | 0.48–1.07 | 0.108 |
| >50000 | 141 (61.0) | 90 (39.0) | Reference | | |

*Note*: OR = Odds Ratio, CI = Confidence interval.

A *p*-value of less than 0.05 was considered statistically significant.

preventing CCHF disease [33]. Moreover, healthcare workers (HCWs) in Baluchistan also had poor knowledge about CCHF [34]. This could be owing to a paucity of skilled staff, essential drugs, and laboratory equipment to deal with CCHF. 72.3% of the population said that CCHF is transmissible and 71% showed adequate knowledge about the symptoms, similar to a study which was conducted in Turkey [35].

Most participants believed that there is an increased risk of people getting the disease during Eid-ul-Adha. In Pakistan, cases are mostly documented sporadically every year, around Eid-ul-Adha. At this time of year, cattle are transported from the countryside to the cities. This allows the CCHF virus to be transmitted through unprotected contact with live animals as well as through contact with animal blood after its slaughter [21, 36, 37].

This study revealed that higher age of participants was significantly associated with positive attitudes, and good practices which affirm with the previous findings [2, 28]. The higher age, the longer is the experience to get familiar with disease, ultimately showing more positive attitudes and practices. The urban population had good knowledge, attitudes, and practices as compared to the rural population in the present study. In Pakistan, the population is comprised of approximately one-third urban (36%) and two-thirds rural (64%) [38]. The majority of people in rural regions who come into direct touch with cattle are illiterate. The majority of the rural people may be unaware of CCHF since they have not encountered a patient in their neighborhood. Moreover, a lack of information and awareness among animal handlers is also responsible for the rapid

**Table 8. Binary logistic regression analysis to find factors of good practices (*N* = 535).**

| Variables | Practice | | OR | (95% CI) | *p*-value |
|---|---|---|---|---|---|
| | **Poor** | **Good** | | | |
| **Gender** | | | | | |
| Male | 170 (56.5) | 131 (43.5) | 0.649 | 0.46–0.92 | 0.014 |
| Female | 107 (45.7) | 127 (54.3) | Reference | | |
| **Age** | | | | | |
| 18–21 | 84 (53.2) | 74 (46.8) | 0.658 | 0.42–1.02 | 0.064 |
| 22–25 | 125 (57.3) | 93 (42.7) | 0.56 | 0.37–0.84 | **0.005** |
| 26 > | 68 (42.8) | 91 (57.2) | Reference | | |
| **Marital status** | | | | | |
| Married | 51 (46.4) | 59 (53.6) | 1.314 | 0.86–2.00 | 0.203 |
| Unmarried | 226 (53.2) | 199 (46.8) | Reference | | |
| **Residence** | | | | | |
| Urban | 218 (50.5) | 214 (49.5) | 1.316 | 0.85–2.03 | 0.214 |
| Rural | 59 (57.3) | 44 (42.7) | Reference | | |
| **Education** | | | | | |
| Primary or below | 2 (40.0) | 3 (60.0) | 0.298 | 0.12–0.85 | 0.058 |
| Matriculation | 18 (69.2) | 8 (30.8) | 0.382 | 0.15–0.95 | **0.040** |
| Intermediate | 53 (58.2) | 38 (41.8) | 0.616 | 0.35–1.08 | 0.093 |
| Graduation | 153 (49.7) | 155 (50.3) | 0.871 | 0.56–1.36 | 0.540 |
| Post-graduation | 52 (49.05) | 54 (50.95) | Reference | | |
| **Monthly income** | | | | | |
| <25000 | 82 (68.9) | 37 (31.1) | 0.403 | 0.25–0.64 | <**0.001** |
| 25000–50000 | 86 (46.5) | 99 (53.5) | 0.998 | 0.69–1.41 | 0.887 |
| >50000 | 108 (47.2) | 122 (52.8) | Reference | | |

*Note*: OR = Odds Ratio, CI = Confidence interval.

A *p*-value of less than 0.05 was considered statistically significant.

spread of CCHF [39, 40]. However, the association between residence either knowledge or attitudes, or practices did not withstand the regression analysis in the current study.

The study showed that well-educated participants had good knowledge, attitudes, and practices as compared to the uneducated or people with lower education which is consistent with the Turkish and Iranian studies [35, 41–43]. This may be reasonable as a higher level of education leads individuals to better attitudes and preventive behaviors regarding CCHF. High levels of education encourage greater research and raise knowledge of cattle diseases. As a result, the average person becomes acquainted with disease preventative measures [44]. Pakistan is a developing country with a very low literacy rate. Due to illiteracy, people don't follow protective measures and even the butchers association denies the presence of the Congo virus in Pakistan to avoid accountability [45].

Our study also shows that the population with high income has good knowledge, attitudes, and practices as compared to the low-income population. This is consistent with many studies that also revealed having a higher socioeconomic position corresponds with having a higher knowledge score [46, 47]. The impact of money or income on health cannot be overstated. Rich individuals are bound to approach data, as well as access better administrations, for example, education when compared to poor individuals [48].

Unfortunately, the Pakistani healthcare system is not prepared to cope with CCHFV epidemics and cannot deal with this significant public health issue for several reasons [49]. These

include the absence of quarantine areas or infection control policies, lack of proper contact tracing procedures, and shortage of trained staff and professionals. So, there is a dire need to increase knowledge among the general population regarding CCHF at all levels by which we can change their attitude toward CCHF. For which the media can play an important role in raising knowledge about the routes of transmission and symptoms of CCHFV, the need of spraying animal folds to protect them from tick attacks, handling and butchering of animals, and the use of proper clothing to reduce contact with ticks in the process of cleaning. An education campaign consisting of seminars, pamphlets, and workshops would be useful in disseminating information, especially in rural areas. There is a strict need to establish quarantine areas and control the migratory activities of people and animals from areas endemic to CCHF, which can prevent the ongoing spread and consequently reduce the number of casualties from CCHF. Level of knowledge and attitude of community is important to get the highest support from community before launching any disease control program [50]. By using these data, policymakers can develop guidelines aimed at addressing the root causes of the rising trend of CCHF in Balochistan province, in particular, and Pakistan in general. The study might be helpful in directing the ministry of health and international organizations to establish and implement a strategic framework for better containment of CCHF and its further spread.

## Limitations

This study has a few limitations. Firstly, the questionnaire distribution was carried out through an online system using different social media platforms. As a result, there is a possibility of bias as underprivileged populations may not have been able to participate in the study. Secondly, in our study, more than half of the participants had a graduation level of education, hence generalization cannot be guaranteed. Participants who do not have any educational degree formed a low proportion of our sample, possibly because they do not have access to social media networks, so future studies may use better approaches to accommodate for this category. A further limitation of the present study is that this study is based on self-reported data, which might contribute to social desirability. It is possible that participants may have answered attitude and practice questions positively based on what they perceive to be expected of them. Nevertheless, the current study was strengthened by a large sample size. To the best of our knowledge, this is the first cross-sectional survey conducted in Pakistan to assess the knowledge, attitudes, and practices towards CCHF among the population of Pakistan, where there is a paucity of literature available. The study will facilitate health officials in the implementation of effective policies to combat the spread of CCHF in Pakistan.

## Conclusions

The findings indicated the lower levels of knowledge, attitudes, and practices of CCHF among Pakistani general people. CCHF is a highly contagious disease that necessitates a comprehensive approach to handle the situation before it spreads further in Pakistan. Preventive interventions are uncommon due to poor infrastructure, a lack of education, and restricted access to health care and livestock-related facilities. It is high time that Pakistan's health, agriculture, and media sectors collaborate with international organizations to establish and implement a strategic framework for CCHF awareness and prevention.

## Acknowledgments

The authors would like to express their heartiest gratitude to all of the study participants for their voluntary involvement.

## Author Contributions

**Conceptualization:** Muhammad Junaid Tahir, Muhammad Saqlain, Ali Ahmed.

**Data curation:** Hashaam Jamil, Muhammad Fazal Ud Din, Muhammad Junaid Tahir, Zair Hassan, Muhammad Arslan Khan, Irfan Ullah.

**Formal analysis:** Muhammad Saqlain.

**Validation:** Muhammad Fazal Ud Din, Muhammad Junaid Tahir, Muhammad Saqlain, Zair Hassan, Muhammad Arslan Khan, Mustafa Sajjad Cheema, Irfan Ullah, Md. Saiful Islam.

**Visualization:** Ali Ahmed.

**Writing – original draft:** Hashaam Jamil, Muhammad Fazal Ud Din, Muhammad Junaid Tahir, Muhammad Arslan Khan, Mustafa Sajjad Cheema, Md. Saiful Islam.

**Writing – review & editing:** Muhammad Saqlain, Zair Hassan, Irfan Ullah, Md. Saiful Islam, Ali Ahmed.

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
