## [Decision Letter · Decision Letter 0]

27 Aug 2022

Dear Mr. Islam,

Thank you very much for submitting your manuscript "Knowledge, attitudes, and practices regarding Crimean-Congo hemorrhagic fever among general people: A cross-sectional study in Pakistan" for consideration at PLOS Neglected Tropical Diseases. As with all papers reviewed by the journal, your manuscript was reviewed by members of the editorial board and by several independent reviewers. In light of the reviews (below this email), we would like to invite the resubmission of a significantly-revised version that takes into account the reviewers' comments. 

We cannot make any decision about publication until we have seen the revised manuscript and your response to the reviewers' comments. Your revised manuscript is also likely to be sent to reviewers for further evaluation.

Sincerely,

Aysegul Taylan Ozkan, M.D., Ph.D.,

Section Editor

Aysegul Taylan Ozkan

Section Editor

Reviewer's Responses to Questions

**Key Review Criteria Required for Acceptance?**

**Methods**

-Are the objectives of the study clearly articulated with a clear testable hypothesis stated?

-Is the study design appropriate to address the stated objectives?

-Is the population clearly described and appropriate for the hypothesis being tested?

-Is the sample size sufficient to ensure adequate power to address the hypothesis being tested?

-Were correct statistical analysis used to support conclusions?

-Are there concerns about ethical or regulatory requirements being met?

Reviewer #1: I would like to thank the Author and colleagues for their effort to share this paper. the methodology is clearly written, although the development of data collection tool and analysis needs some discussion particularly when we are considering cutoff points and questions related to practice. Details will be explained in general comments.

Reviewer #2: The authors may include the sampling formula indicate in the online source they mentioned in methods section.

**Results**

-Does the analysis presented match the analysis plan?

-Are the results clearly and completely presented?

-Are the figures (Tables, Images) of sufficient quality for clarity?

Reviewer #1: the way that the data analysis has been presented might be misleading for both participants in the research questionnaire and reader of the paper. I would like to share with you some points in general comments sections.

Reviewer #2: The authors should also mention why and how they divided the margins of good/poor level of knowledge with references (if there is any). The discussion sections need some incorporation of plausible explanations of the findings as well.

**Conclusions**

-Are the conclusions supported by the data presented?

-Are the limitations of analysis clearly described?

-Do the authors discuss how these data can be helpful to advance our understanding of the topic under study?

-Is public health relevance addressed?

Reviewer #1: Discussion and conclusion sections are written simply even with new analysis some points might raise up for deeper discussion.

Reviewer #2: Although the authors described its novelty, they should describe the broader implications of this study and how it may contribute to the scientific community apart from Pakistan. Another shortcoming of the study is its language.

**Editorial and Data Presentation Modifications?**

Reviewer #1: (No Response)

Reviewer #2: Major revision.

**Summary and General Comments**

Reviewer #1: I would like to thank the authorship team for their effort in developing this paper. Also, I would like to share with you some points for more clarification / change to better understanding of the content of the paper.

- In the abstract and results sections, it is confusing to say only 51.5%, around half is not only. Please consider this.

- In the background section. the magnitude of the disease "CCHFV" is not clear globally or in Pakistan. Please elaborate more about the situation in Pakistan particularly after 2020 as you mentioned in discussion.

- Culturally, I think it would be a good idea to explain what is Eid ul Adha and what is qurban with high number of animal slaughtering in a short time.

- in sampling size section, I assume that the 1039 is a number of completed surveys please correct me if not and please add completeness % if possible and how did you manage missing data.

- in the Questionnaire development section, please explain how did you identify cutoff points in each component (Knowledge, Attitude, Practice)? and do you think it is high in knowledge? 

- In Results section, I would like to recommend to classify the table 1 according to Have you heard about Congo fever? "in columns against other values. 

- Also, for age ranges are very narrow, Why?

-most of participants are in urban area, does this explain low knowledge level?

- in table 2, As you choose to use the questions written in the survey, please add correct response (eg: Congo fever disease was first characterized in which area? Correct "XXX")

- many questions in Attitudes and practices analysis part in not related to the same section. please clarify For example "Do you think there is a need of more awareness of CCHF in general public about CCHF" not asking about the practice. 

- in the Attitudes and practices sections, questions are very directed particularly when you ask about source of knowledge and following the right procedures. (Everyone will say yes!!) my recommendation is to clarify if it is different in Urdu or to discuss this point deeply and put in limitation section. this point is very important for questionnaire validation. 

- regarding to limitation section. I would prefer to ignore recall bias as it is very minimal in this study and consider more the limitation in the data collection tool.

Thank you again for this paper and wishing you all the best.

Reviewer #2: Thank you for sending me through this paper on Knowledge, attitude and practices regarding Crimean-Congo hemorrhagic fever among general people: A cross-sectional study in Pakistan. I have read it thoroughly and noted the following observations regarding the manuscript. 

The introduction section is well-defined with adequate references. The authors may include the sampling formula indicate in the online source they mentioned in methods section. The authors should also mention why and how they divided the margins of good/poor level of knowledge with references (if there is any). The discussion sections need some incorporation of plausible explanations of the findings as well. Although the authors described its novelty, they should describe the broader implications of this study and how it may contribute to the scientific community apart from Pakistan. Another shortcoming of the study is its language. The author(s) may also get the manuscript revised by someone who excels in English language to enhance its quality. Thank you.

PLOS authors have the option to publish the peer review history of their article (what does this mean?). If published, this will include your full peer review and any attached files.

Reviewer #1: Yes: Mohamed A. Abdelbaqy

Reviewer #2: No
---

## [Decision Letter · Decision Letter 1]

15 Oct 2022

Dear Mr. Islam,

Thank you very much for submitting your manuscript "Knowledge, attitudes, and practices regarding Crimean-Congo hemorrhagic fever among general people: A cross-sectional study in Pakistan" for consideration at PLOS Neglected Tropical Diseases. As with all papers reviewed by the journal, your manuscript was reviewed by members of the editorial board and by several independent reviewers. In light of the reviews (below this email), we would like to invite the resubmission of a significantly-revised version that takes into account the reviewers' comments. 

We cannot make any decision about publication until we have seen the revised manuscript and your response to the reviewers' comments. Your revised manuscript is also likely to be sent to reviewers for further evaluation.

Sincerely,

Aysegul Taylan Ozkan, M.D., Ph.D.,

Section Editor

Aysegul Taylan Ozkan

Section Editor

Reviewer's Responses to Questions

**Key Review Criteria Required for Acceptance?**

**Methods**

-Are the objectives of the study clearly articulated with a clear testable hypothesis stated?

-Is the study design appropriate to address the stated objectives?

-Is the population clearly described and appropriate for the hypothesis being tested?

-Is the sample size sufficient to ensure adequate power to address the hypothesis being tested?

-Were correct statistical analysis used to support conclusions?

-Are there concerns about ethical or regulatory requirements being met?

Reviewer #1: - The authors are doing a lot to make the paper more informative and easier to be understood for the reader. The study design is answering the objectives clearly.

- The sampling technique is good, But I would like to highlight some point to be changed in the next version.

1) The author mentioned that "... and the greater the generalizability of the study"

Comment: The convenient sampling technique is not for generalization, where there is no randomization and also because of that the internet access in Pakistan was 25% of total population in 2020. 

Recommendation: Delete this sentence.

2) The cut of points for KAP for each is not clear how it was calculated. I mean that for Knowledge why 9 not 8 or 10. and it is the same for attitude and practice scores. 

Recommendation: If the author have a solid reason for that, (S)he can explain more, otherwise it will be better to use median or quartile or quantile to describe the results. 

3) in "Statistical analysis" part the author mentioned that "Descriptive statistics were applied as means and standard deviations for continuous variables...." while in analysis there is no use of mean or SD. 

Recommendation: Remove this sentence.

Reviewer #2: See general comments/comment to the editor.

**Results**

-Does the analysis presented match the analysis plan?

-Are the results clearly and completely presented?

-Are the figures (Tables, Images) of sufficient quality for clarity?

Reviewer #1: The results are presented very well and clear with alot of effort to get appropriate information from the data. the following are some comments to be answered in the next version.

1) the following question are not related to practice. 

- Do you think keeping livestock at home poses you an extra risk of getting CCHF

- Do you think there is a need of more awareness of CCHF in general public about CCHF

- Do you think hospitals in your area are provided with sufficient facilities to diagnose and treat CCHF

Recommendation: Eighter to relocate those questions or merge the both tables (Attitude and practice) and analyse them as one component.

Reviewer #2: See general comments/comment to the editor.

**Conclusions**

-Are the conclusions supported by the data presented?

-Are the limitations of analysis clearly described?

-Do the authors discuss how these data can be helpful to advance our understanding of the topic under study?

-Is public health relevance addressed?

Reviewer #1: (No Response)

Reviewer #2: See general comments/comment to the editor.

**Editorial and Data Presentation Modifications?**

Reviewer #1: The overall of the paper is attractive and smoothly ongoing from the title to the reference section. Also, the authors try to follow the comments in that revision and discussed them all in a noticeably clear way. Thank you.

Just small comment "Attitudes and practices" section, after table 4; the author mentioned that "The majority of participants 40 .9% (n=219) strongly agreed that there... ", actually we cannot say majority where the percentage is 40%. Please review

Reviewer #2: See general comments/comment to the editor.

**Summary and General Comments**

Reviewer #1: A good scientific effort is done in this paper, and I would like to congratulate the authors for the paper and to thank them for replying the previous version comments

Reviewer #2: I have read the paper on Knowledge, attitudes, and practices regarding Crimean-Congo hemorrhagic fever thoroughly and noted the following observations regarding the manuscript. 

The paper is written well and presents some interesting insights on a relatively neglected field of knowledge. The background section is well-defined with adequate references. However, the authors should mention why they chose convenient sampling method when the study was conducted on general population. Although they mentioned they used digital platforms which was more accessed by graduate level people in the limitations, they referred to further studies instead for providing justification for how they addressed the shortcomings. Apart from that, the results and the discussion sections are described properly. I encourage the authors to address these minor issues, and wish them all the best. Thank you.

PLOS authors have the option to publish the peer review history of their article (what does this mean?). If published, this will include your full peer review and any attached files.

Reviewer #1: Yes: Mohamed Abdelbaqy

Reviewer #2: No
---

## [Decision Letter · Decision Letter 2]

22 Nov 2022

Dear Mr. Islam,

Thank you very much for submitting your manuscript "Knowledge, attitudes, and practices regarding Crimean-Congo hemorrhagic fever among general people: A cross-sectional study in Pakistan" for consideration at PLOS Neglected Tropical Diseases. As with all papers reviewed by the journal, your manuscript was reviewed by members of the editorial board and by several independent reviewers. The reviewers appreciated the attention to an important topic. Based on the reviews, we are likely to accept this manuscript for publication, providing that you modify the manuscript according to the review recommendations. 

Sincerely,

Aysegul Taylan Ozkan, M.D., Ph.D.,

Section Editor

Aysegul Taylan Ozkan

Section Editor

Reviewer's Responses to Questions

**Key Review Criteria Required for Acceptance?**

**Methods**

-Are the objectives of the study clearly articulated with a clear testable hypothesis stated?

-Is the study design appropriate to address the stated objectives?

-Is the population clearly described and appropriate for the hypothesis being tested?

-Is the sample size sufficient to ensure adequate power to address the hypothesis being tested?

-Were correct statistical analysis used to support conclusions?

-Are there concerns about ethical or regulatory requirements being met?

Reviewer #1: (No Response)

Reviewer #2: The methods section looks well-described. The objectives of the study is clearly articulated, and the authors used appropriate study design. The sample size, hypothesis test and the analysis done are appropriate. However, the authors should mention how they constructed the questionnaire. They should provide references if they used any, or just state if they constructed an inventory on their own. Providing references would strengthen how they labeled any knowledge poor or an attitude positive/negative.

**Results**

-Does the analysis presented match the analysis plan?

-Are the results clearly and completely presented?

-Are the figures (Tables, Images) of sufficient quality for clarity?

Reviewer #1: (No Response)

Reviewer #2: The results clearly and completely presented. The analysis presented also look standard.

**Conclusions**

-Are the conclusions supported by the data presented?

-Are the limitations of analysis clearly described?

-Do the authors discuss how these data can be helpful to advance our understanding of the topic under study?

-Is public health relevance addressed?

Reviewer #1: (No Response)

Reviewer #2: The authors may consider adding a few sentences of recommendations based on the specific findings from binary logistic regression analysis. For example, what are the implications of the association found between education and income status and the KAP of the participants. More specifically innicate what needs to be done.

**Editorial and Data Presentation Modifications?**

Reviewer #1: (No Response)

Reviewer #2: The paper presents interesting findings on a novel topic. The paper can be considered for publication with a few modifications as suggested. I suggest a “Minor Revision”.

**Summary and General Comments**

Reviewer #1: I would like to thank the authors for their valuable work and scientific discussion they have with this paper, also for clarifying the previous comments. A few points would like to discuss as following

- Most of the previous points were discussed clearly.

- Regarding the "cut-off points", what you mentioned as a response for the previous review round sounds very clear. The point is to share with readers of the journals. So, you can explain how the review research committee decide that and how you follow the recommendations of review research committee. 

- Regarding the "Attitude and practice" part, it is totally understandable the point of culture variation and the translation from language to another. Yes some verbs can give behaviour meaning in language and attitude meaning in another language. but here the situation is totally different, the language of manuscript is English and the reader might not get it, please be sure that there is no overlap between attitude and practice in the manuscript. 

Fo example: the following questions you mentioned in manuscript

 1- Do you think keeping livestock at home poses you an extra risk of getting CCHF

 2- Do you think there is a need of more awareness of CCHF in general public about CCHF

 3- Do you think hospitals in your area are provided with sufficient facilities to diagnose and treat CCHF

how you can clarify "do you think" to action??!! 

As a recommendation please merge the attitude and practice as one part, OR, review the translation of "do you think" in Urdu is a practice verb, Or remove this part, and of course reflect to regression 

At the end I would like to thank you very much for the manuscript and the fruitful information on it.

Best regards

Reviewer #2: The paper presents interesting findings on a novel topic. The paper can be considered for publication with a few modifications as suggested.

PLOS authors have the option to publish the peer review history of their article (what does this mean?). If published, this will include your full peer review and any attached files.

Reviewer #1: Yes: Mohamed Abdelbaqy

Reviewer #2: No

Figure Files:

Data Requirements:

Reproducibility:

References

---

## [Editor Report · Decision Letter 3]

28 Nov 2022

Dear Mr. Islam,

We are pleased to inform you that your manuscript 'Knowledge, attitudes, and practices regarding Crimean-Congo hemorrhagic fever among general people: A cross-sectional study in Pakistan' has been provisionally accepted for publication in PLOS Neglected Tropical Diseases.

Best regards,

Aysegul Taylan Ozkan, M.D., Ph.D.,

Section Editor

Aysegul Taylan Ozkan

Section Editor

---

## [Editor Report · Acceptance letter]

5 Dec 2022

Dear Mr. Islam,

We are delighted to inform you that your manuscript, "Knowledge, attitudes, and practices regarding Crimean-Congo hemorrhagic fever among general people: A cross-sectional study in Pakistan," has been formally accepted for publication in PLOS Neglected Tropical Diseases.

Best regards,

Shaden Kamhawi

co-Editor-in-Chief

Paul Brindley

co-Editor-in-Chief
